# Cloning of maize *chitinase* 1 gene and its expression in genetically transformed rice to confer resistance against rice blast caused by *Pyricularia oryzae*

Sadaf Anwaar[1☯], Nyla Jabeen[1☯], Khawaja Shafique Ahmad[2]*, Saima Shafique[3], Samra Irum[1], Hammad Ismail[4], Siffat Ullah Khan[5], Ateeq Tahir[6], Nasir Mehmood[7], Mark L. Gleason[8]

1 Department of Biological Sciences, International Islamic University, Islamabad, Pakistan, 2 Department of Botany, University of Poonch Rawalakot, Rawalakot, Azad Jammu and Kashmir, Pakistan, 3 Department of Plant Breeding and Molecular Genetics, University of Poonch Rawalakot, Rawalakot, Azad Jammu and Kashmir, Pakistan, 4 Department of Biochemistry and Biotechnology, University of Gujrat, Gujrat, Pakistan, 5 Biotechnology Research Institute, Chinese Academy of Agricultural Sciences, Beijing, China, 6 Institute of Agricultural Sciences, University of the Punjab, Lahore, Pakistan, 7 Department of Botany, Rawalpindi Women University, Rawalpindi, Pakistan, 8 Department of Plant Pathology and Microbiology, Iowa State University, Ames, Iowa, United States of America

☯ These authors contributed equally to this work.
* shafiquebot@gmail.com, ahmadks@upr.edu.pk

**Data Availability Statement:** All relevant data are within the paper and its Supporting information files.

## Abstract

Fungal pathogens are one of the major reasons for biotic stress on rice (*Oryza sativa* L.), causing severe productivity losses every year. Breeding for host resistance is a mainstay of rice disease management, but conventional development of commercial resistant varieties is often slow. In contrast, the development of disease resistance by targeted genome manipulation has the potential to deliver resistant varieties more rapidly. The present study reports the first cloning of a synthetic maize *chitinase* 1 gene and its insertion in rice cv. (Basmati 385) via *Agrobacterium*-mediated transformation to confer resistance to the rice blast pathogen, *Pyricularia oryzae*. Several factors for transformation were optimized; we found that 4-week-old calli and an infection time of 15 minutes with *Agrobacterium* before colonization on co-cultivation media were the best-suited conditions. Moreover, 300 µM of acetosyringone in co-cultivation media for two days was exceptional in achieving the highest callus transformation frequency. Transgenic lines were analyzed using molecular and functional techniques. Successful integration of the gene into rice lines was confirmed by polymerase chain reaction with primer sets specific to *chitinase* and *hpt* genes. Furthermore, real-time PCR analysis of transformants indicated a strong association between transgene expression and elevated levels of resistance to rice blast. Functional validation of the integrated gene was performed by a detached leaf bioassay, which validated the efficacy of chitinase-mediated resistance in all transgenic Basmati 385 plants with variable levels of enhanced resistance against the *P. oryzae*. We concluded that overexpression of the maize *chitinase* 1 gene in Basmati 385 improved resistance against the pathogen. These findings will add new options to resistant germplasm resources for disease resistance breeding. The maize

**Funding:** The study was supported by Higher Education Commission of Pakistan under grant number 8958, however funders had no role in study design, data collection and analysis, decision to publish, or preparation of the manuscript.

**Competing interests:** The authors have declared that no competing interests exist.

*chitinase* 1 gene demonstrated potential for genetic improvement of rice varieties against biotic stresses in future transformation programs.

## Introduction

Yield loss due to diseases and pests has been estimated at >10% for crops worldwide [1]. Rice (*Oryza sativa* L.) is a predominant human food source and is widely grown worldwide, especially in Asia and Africa [2]. Rice provides 20% of the dietary energy for the world's population [3]. In Pakistan, rice is the second most important food crop after wheat. Akey export product [4, 5], it is grown on 4.2 million hectares. Pakistan is the third largest exporter and 12th largest producer of rice in the world [6], and the crop accounts for 3% of value-added agricultural returns and 0.6% of Pakistan's gross domestic product [7].

In Pakistan, fungal diseases of rice cause the most severe yield losses [8]. Rice blast, caused by the fungus *Pyricularia oryzae*, is one of the most damaging fungal diseases, causing heavy losses in quality and yield [9]. Many basmati rice varieties grown in Pakistan are susceptible to blast [10].

Resistance breeding and pesticides are the two most commonly used methods to combat plant diseases. The extensive use of pesticides leads to severe environmental problems [11]. Whereas the continuous use of traditional breeding methods may limit the gene pool of a species from which cultivars are derived, making crops more susceptible to biotic and abiotic stresses, and impeding future growth [12]. The world's population is projected to rise from 6 to 8 billion by 2020, with rice consumption expected to grow by 1.8 percent annually. As a result, rice production must be increased by 25–45 percent to meet the increasing demand [12]. Efforts are being made to develop rice to produce high-quality crops [13] genetically. The most effective alternative is genetically engineered plants with enhanced characteristics [14].

One of the most effective strategies to defend rice against blast is development and cultivation of resistant genotypes [15]. The genomes of plants, fungi, and insects encode several types of chitinase, a chitin-degrading glycosidase. These genes have been comprehensively studied owing to their significant role in plant defense responses against fungal pathogens [16, 17]. Thirty-three *chitinase* genes were characterized in maize (*Zea mays*) genome codes for producing the chitinase-like protein present in a quantitative trait locus for fungal maize ear rot resistance [18]. These studies served as a platform to use the maize *chitinase* gene for developing transgenic rice with fungal resistance. During the past two decades, a wide range of cloned genes, including chitinase, has been utilized to confer resistance against fungal pathogens in plants [19, 20].

Stable genetic transformation of cereal crops like maize, rice, wheat, and barley has provided new and improved varieties grown in extensive planting throughout the world [21]. Numerous methods for transferring genes into rice have been developed [22]. However, *Agrobacterium*-mediated methods have been widely used because of high transformation efficiency and the inclusion of transgenes with single or low copy numbers or large DNA segments with distinct ends that can be integrated readily into host genomes [23].

In view of the significance of rice in global food security and its economic importance in Pakistan, the objective of the current research was to develop a transgenic rice variety with enhanced resistance against the fungal pathogen *Pyricularia oryzae* by introducing the maize *chitinase* 1 gene through *Agrobacterium*-mediated transfer.

## Materials and methods

### Cloning of maize *chitinase* 1 gene

Cloning of the maize *chitinase* 1 gene (*MCI*) was carried out in the Department of Biotechnology, Plant Transformation Facility, Iowa State University (ISU), Ames, Iowa, USA. The maize *chitinase* 1 gene (accession number NP_001148230.1) was retrieved from the NCBI database. Based on its sequence homology with maize gene encoding class I *chitinase* protein, its codon was optimized according to rice using the codon optimization tool. The gene was synthesized from Gene Universal USA and was provided in standard vector pUC 57. The vector pDW3586 and all the protocols followed during this part of the research were provided by Dr. David A. Wright, ISU, with slight modifications (Fig 1); the resulting vector was named pSDF1.

**Competent cell formation, plasmid extraction and restriction digestion.** *E. coli* strain DH 10B was transformed with pSDF1. The plasmid was extracted using an IBI Scientific plasmid extraction kit and digested with two pairs of endonucleases, Xba I/SpeI and XhoI/BamHI. The digested plasmids were then run on a 0.8% gel with 2 log markers. Expected bands were 9.1 and 6kb and 13.4 and 1.8 kb, respectively. The desired plasmid band pSDF1 was purified using an IBI Scientific gel purification kit, and the DNA was sent to the ISU DNA Facility for sequencing.

After sequence confirmation of vector pSDF 1, we transformed the DH 10B strain with pSDF2. For convenience, standard vector pUC 57(with *MC 1*) gene was renamed pSDF2. The gene was obtained from pSDF2, integrated into pSDF1, and the product was termed pSDF3. The restriction digestion of plasmid pSDF1 and pSDF2 was done with endonucleases Spe I and Bam HI. Digested plasmids were run on a 0.8% gel. A 10-kb fragment was produced after restriction digestion of pSDF1 whereas a 0.9-kb maize *chitinase* 1 gene fragment was produced after restriction digestion of pSDF2. For the ligation step, vector (pSDF1), *MC1* gene, T4 DNA ligase Buffer and T4 DNA ligase were mixed and incubated overnight at 16 C.

The next step was to transform *E. coli* with plasmid pSDF 3. Plasmid extraction was done using an IBI Scientific kit and was digested with endonuclease pair SpeI and BamHI, then run on a 0.8% agarose gel to confirm ligation. After confirmation of ligation, pSDF3 was sequenced at the ISU DNA Facility. The sequencing confirmed the results of the cloning experiment.

We then transformed a fresh culture of *Agrobacterium tumefaciens* strain EHA 101 with vector pSDF 3. The culture was centrifuged, and the pellet was washed three times with cold nuclease-free water. For *Agrobacterium* cells to lyse, 5 ul of PD2 buffer from the IBI Scientific plasmid extraction kit was added to a microcentrifuge tube. The tube was then placed into a thermocycler for incubation for 5 minutes at 60 C in order to lyse *Agrobacterium*. The plasmid DNA that was obtained was then used in competent cell formation of *E. coli*, and extraction of plasmid and digestion with the same pair of endonucleases SpeI and BamHI was performed. Afterward, the digested plasmid was run on agarose gel electrophoresis to confirm the successful transfer of pSDF3 (Fig 2) to *Agrobacterium*.

### Generation of transgenic rice lines via *Agrobacterium*-mediated gene transformation

*Oryza sativa* L. indica cultivar Basmati 385 seeds were used as explants in the research work. Seeds were obtained from the National Agriculture Research Centre (NARC), Pakistan.

**Agrobacterium strain and culture conditions.** Five ml of a primary culture of fresh *Agrobacterium* was utilized to inoculate 50 ml of LB broth media to grow the *Agrobacterium* to an OD600 of 0.5–1.0. The culture was centrifuged at 8000 rpm for 3 minutes to form a pellet and resuspended in MS liquid (MSL) media before infection. The vector used in the study had

**Fig 1. Detailed map of vector pDW3586.**

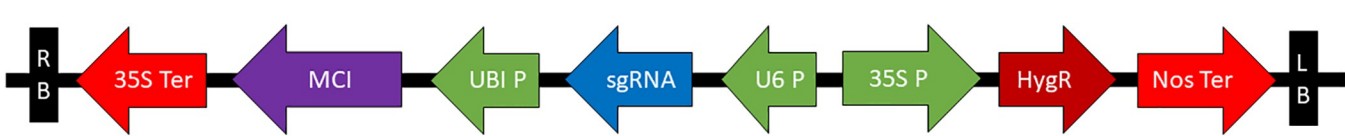

**Fig 2. T-DNA region of plasmid pSDF3.**

**Table 1. Media formulations used in rice transformation.**

| Media | Duration | Components |
|---|---|---|
| Callogenesis | 12–15 days | N6 salt, 2,4-D (2 mg/L), CuO NPs (10mg/L), Sucrose 30 g/L, Phytagel 8 g/L, pH 5.8. |
| Bacterial suspension | 2 days | LB medium, sucrose 3%, Acetosyringone 200 μM, pH 7. |
| Infection | 5-25minutes | MS salt, sucrose 3%, pH 5.8. (O.D$_{600}$ of 0.1–0.2) |
| Co-cultivation | 1–3 days | MS salt, 2,4-D 2 mg/L, Sucrose 3%, Acetosyringone (100μM-400 μM), Phytagel 10g/L, pH 5.8. |
| Washing | 2–3 minutes | MS salt, sucrose 30 g/L,Carbenicillin 400mg/L, pH 5.8. |
| Selection | 3–4 weeks | MS salt, 2,4-D 2 mg/l, SMI (50 mg/L Hygro) and SMII (50 mg/L Hygro+400 Carbenicillin), Phytagel 8g/L, pH 5.8 |
| Regeneration | 5–8 weeks | MS salt + (NAA 1.0 mg/L + BAP 0.5 mg/L + Kin 0.5mg/L), Sucrose 3%, Phytagel 16g/L, pH 5.8, RGMI (Hygro 25 mg/L, Carb. 400 mg/L) and RGM II (Hygro 25 mg/L, Carb. 400 mg/L, CuO- NPs 20 mg/L) |

hygromycin as the selection marker gene; thus, calli were studied for their capacity to tolerate hygromycin. Calli were grown for four weeks on agar media at four different concentrations (25, 50, 75, and 100 mg/L) of hygromycin.

*Agrobacterium* **infection and co-cultivation.** Four-week-old calli were immersed in MSL media with suspended *Agrobacterium*, then diluted to 0.1 to 0.2 at OD600. Infection time varied from 5 to 25 minutes, with occasional shaking. The Calli were then dried on sterilized filter paper to eliminate excess bacteria and were shifted to co-cultivation media plates. Co-cultivation media with different acetosyringone concentrations (100, 200, 300, and 400 μM) were tested. The dried infected calli were exposed to two, three or four days of co-cultivation at 22°C in darkness [24]. Excess *Agrobacterium* was eliminated after co-cultivation by washing the calli with the antibiotic carbenicillin. Calli were washed with sterilized distilled water and then with MS liquid containing carbenicillin (400mg/L) for 10 minutes [25].

**Selection of transformed calli and regeneration of transformed plantlets.** After washing, selection of transformed calli was made on two different media (Table 1). Petri plates containing calli on selection media were kept in a growth chamber for 2 to 3 weeks under 16 hrs light /8 hrs darkness at 3°C. The hygromycin-resistant calli were estimated, and the precent efficiency of callus transformation efficiency on the selection media was calculated. We followed the transformation protocol reported in [24] with slight modifications.

Hygromycin-resistant calli were shifted to two different regeneration media, RGM I (hygro 25 mg/L+ Carbenicillin 400 mg/L) and RGM II (Hygromycin 25 mg/L, carbenicillin 400 mg/ L, CuO-NPs 20 mg/L). The calli were incubated in a growth chamber under 16 hrs light /8 hrs darkness at 28°C for four weeks. Regeneration frequency was calculated by percentage. The plants thus obtained from regeneration were subjected to molecular analysis through PCR to confirm the successful integration of *MC1* gene as documented by [26] with slight modifications. The media formulations used in the transformation process are listed in Table 1.

## Molecular analysis of putative transgenic rice

Basmati 385 was subjected to molecular analysis to confirm transformation. For PCR, DNA was isolated from transformants using an Invitrogen Genomic Plant DNA Purification kit, and plasmid DNA was isolated from *Agrobacterium* strain EHA 101 with an IBI Scientific DNA extraction kit. The protocol followed with modifications for PCR was reported

previously [26]. The hygromycin phosphotransferase gene (*hpt*) and maize *chitinase* 1 gene in putative transformants were amplified using Thermo Fisher PCR master mix with the following primer pairs:

**hpt gene** FP-ACAGCGTCTCCGACC TGATGCA

RP-AGTCAATGACCGCTGTTATGCG

**MC1 gene** FP- GGTAAGGTCTTGGCATCATTTA

RP-TGGCCCTCTATTTCGTACTTGAAC

Primer sequences for *actin* gene were used as an internal control, The amplification reaction was performed with denaturation at 95˚C for 2 minutes, annealing of primers at 52˚C for 1 minute, and extension at 72˚C for 2 minutes. The cycle was repeated 29 times to give exponential amplification of the desired DNA sequence and final extension at 72˚C for 5 min.

**Expressional analysis of *MC1* gene in Basmati 385.**   After the MC1 and hpt genes were confirmed in transgenic rice, expression of MC1 was analyzed using real-time PCR at both T0 and T1 generations and at seedling and tillering stages. Total RNA extraction was done with the method of Liu et al. [27] with slight modifications and cDNA was synthesized using Thermo Fischer Revert-Aid First Strand cDNA Synthesis Kit K-1622. The analysis was performed on a Bio-Rad Real-Time PCR system, using the transformed cDNA as templates and *MC1* primers to analyze the gene. The reference gene was actin (*ACT 1*) with to following primer sequence:

FP -CTTCATAGGAATGGAAGCTGCGG

RP -CGACCACCTTGATCTTCATGCTGCTA

To quantify the amount of dsDNA, One-Step SYBR Green Master Mix was used. The thermal setting was denaturation at 95˚C for 3 minutes, followed by 39 cycles of denaturation at 94˚C for 30 seconds, annealing at 52˚C for 30 seconds and extension at 72˚C for 30 seconds. The qPCR samples were run in sets of three. Actin was utilized as an internal control for data standardization. Comparative quantification of gene expression was analyzed using the delta Ct method in tested samples and the standard deviation was calculated [28]. The graphs were plotted for the relative expression of *MC1* gene compared to the internal control.

## Bioassay to evaluate transgenic rice response against *P. oryzae*

A detached leaf bioassay was carried out on control and transgenic plants to evaluate the phenotypic response of transgenic plants caused by *MC1* against *Pyricularia oryzae*. For this purpose, the mycelium of *P. oryzae* was used for the inoculum preparation on PDA plates at 28˚C for 48 hrs. Precisely for inoculum preparation, 10 drops of Tween 20 were added in 10 ml of water in a beaker and mixed it well by stirring. Few drops of distilled water were placed on a fungal plate using a disinfected spatula. The mixture was filtered, and spores were quantified using a hemocytometer. The conidial spore suspension was adjusted to 5x100 spores/ml in distilled water as described. This spore suspension was inoculated on the adaxial surface of detached leaves (transgenic and control plant) and transferred onto moist filter paper in glass petri plates were sealed with parafilm to retain the moisture [19]. Lesion size (length × maximum width) was measured after one week of inoculation at 28˚C. The severity of blast symptoms on each leaf was rated using a qualitative scale [28] as follows: 0 = no visible lesion on the leaf, 1 = up to 10% leaf area affected, 2 = 11–25% leaf area affected, 3 = 26–50% leaf area affected, 4 = 51–75% leaf area affected, 5 = more than 75% leaf area affected. Disease

severity ratings were converted into percent diseased tissue using the formula [29].

$$PBI = \frac{sum\ of\ all\ ratings}{No.\ of\ leaves\ X\ maximum\ disease\ score}$$

## Results

### Cloning of maize *chitinase 1* gene

*E. coli* strain DH 10 was transformed with pSDF1, grown in LB media amended with 100 mg/L of streptomycin. The plasmid (pDW3586) was 17.34 kb and digested with two pairs of endonucleases (Xba I and Spe I) loaded in lanes D1-D3 where D1, D2 and D3 represent restriction incubation time for 3 h, 6 h and 12 h respectively. Bands of 9.1 kb and 6kb were expected from Xba I and Spe I restriction, whereas for Xho I and Bam HI, bands of 13.4 kb and 1.8 kb were expected. From left D4 gave expected bands of 13.4 and 1.8 kb and was selected for purification and sequencing. Results of competent cell formation and plasmid digestion are shown in Fig 3.

After sequence confirmation of pSDF1, pSDF 2 (Fig 3a and 3b) carrying *MC1* produced two bands—2.7 kb (pUC 57 gene) and 0.9 kb (maize *chitinase* 1 gene). The digest in lane 2 showed brighter bands and was subjected to gel clean-up and DNA extraction (Fig 3d).

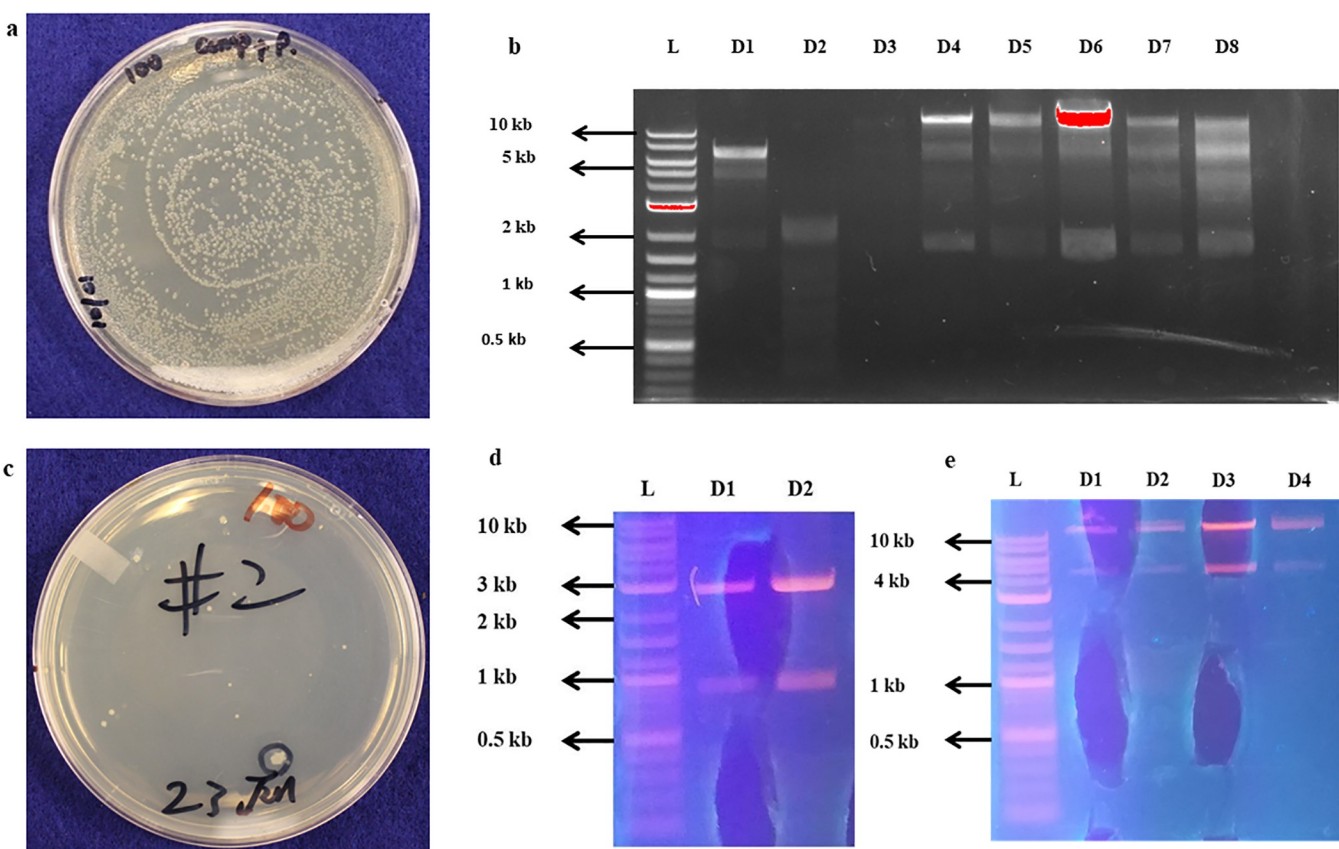

**Fig 3. (a) Competent cells of pSDF1, (b) Restriction digestion of pSDF1 with XhoI / BamHI, (c): Competent cells of pSDF2, (d) Restriction Digestion of pSDF2 BamHI / SpeI, I (e) Restriction Digestion of pSDF1 BamHI / SpeI.**

Moreover, pSDF1 was digested with the same endonucleases and run-on gel to get the desired vector (Fig 3e) for ligation of *MC1*.

The bands of digested pSDF 1 (from D3) and MC1 (from D2) were ligated to form pSDF 3, the desired plasmid carrying our gene of interest. To confirm successful ligation of *MC1* in pSDF1, the plasmid was cloned in *E. coli* and digested with SpeI/Bam HI. The results of competent cell formation and digestion of pSDF3 are shown in (Fig 4a). The gel picture (Fig 4b) clearly showed the presence of both plasmid (10 kb) and *MC1* (0.9 kb) bands. The plasmid digest in lane 3 (D3) showed the brightest band and was selected for sequencing. Sequencing confirmed the cloning results. When the sequence of pSDF3 was confirmed by Sanger sequencing, it was transformed into *Agrobacterium* strain EHA 101, competent cell formation (Fig 4c) and restriction digestion of the plasmid with the same pair of endonucleases was done (Fig 4d).

### *Agrobacterium*-mediated transformation of Basmati 385 with *MC1* gene

Table 2 represents the effects of different hygromycin concentrations on the proliferation and growth of calli. Of all the tested concentrations of hygromycin, 25 mg/L was best suited, with 82% of calli showed normal growth and proliferation. At 50 mg/L, the calli showed negligible growth and 62% turned brown, whereas oat 75 mg/L, only 5% of calli showed growth, 45% turned brown, and 50% of calli died. At a concentration of 100 mg/L, all of the calli were dead.

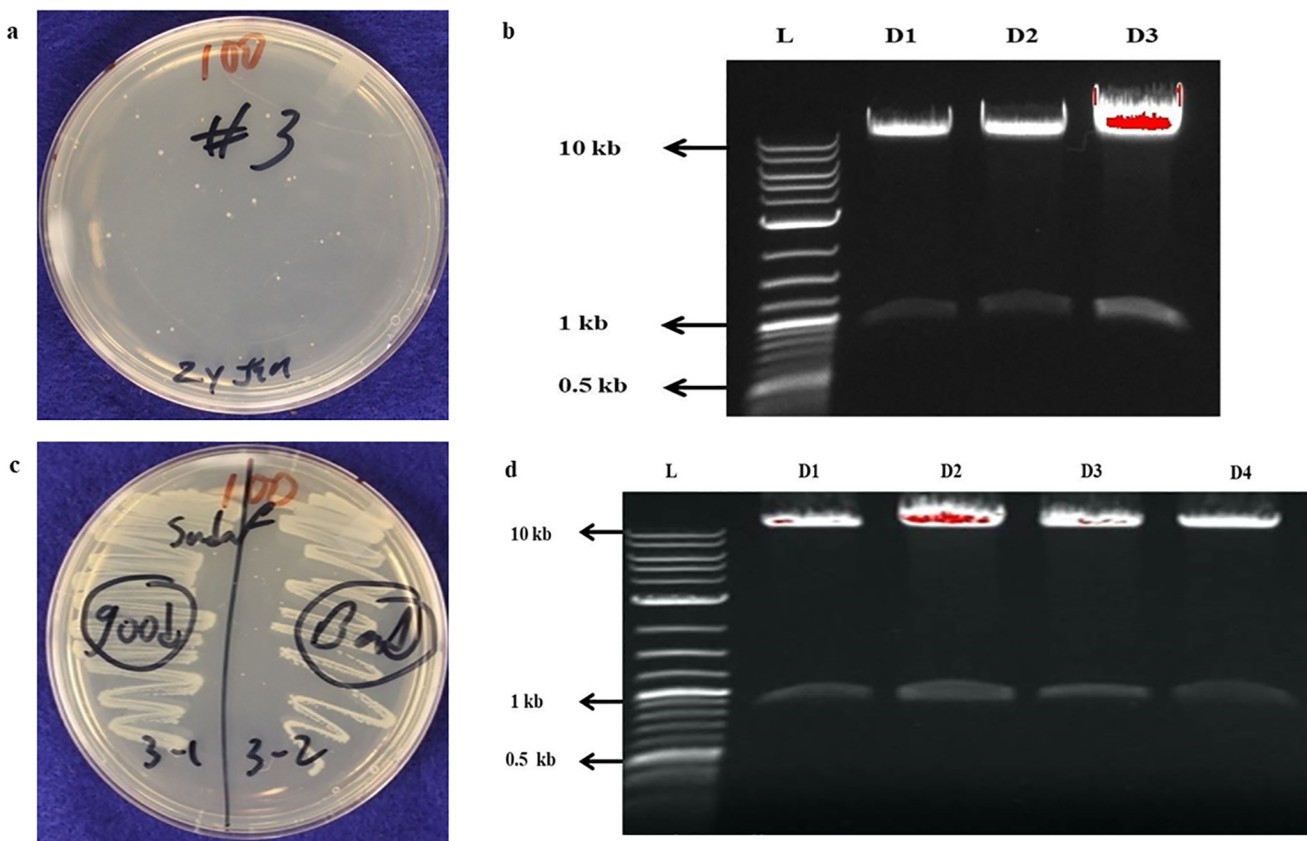

**Fig 4. (a) Competent cells of pSDF3 (b) Confirmation of ligation of *MC1* gene in pSDF3 through restriction digestion c): Competent cells of *Agrobacterium* strain EHA 101 with pSDF3 (d) Confirmation of transformation of *Agro* strain with pSDF3 through restriction digestion.**

**Table 2. Determination of the calli sensitivity to different concentrations of hygromycin.**

| Conc. of hygromycin (mg/L) | %of calli turned brown | % of calli turned dead | % of calli showing proliferation | Results |
|---|---|---|---|---|
| 25 | 18 | 0 | 82 | Strongly proliferating calli |
| 50 | 62 | 18 | 20 | H = Negligible growth of calli |
| 75 | 45 | 50 | 05 | Majority of calli died |
| 100 | 0 | 100 | 0 | All calli died |

No of calli inoculated = 50 per concentration

Age of calli and infection time with agro strain is considered vital. The results of these factors are documented in Table 3 Four-week-old calli with infection time of 15 minutes showed the highest incidence of regeneration (59%). In comparison, 2- and 6-week-old calli with 15 minutes of infection time exhibited 36% and 33% regeneration, respectively.

The highest response, 64%, was recorded after a co-cultivation period of 2 days with 300 µM/L of acetosyringone, followed by 53% on 200 µM/L and 23% on both 100 and 400 µM/L (Fig 4). When the inoculation time increased to 3 days, a significant decrease in the regeneration of calli was observed; maximum regeneration was recorded at 33% on 300 µM/L of acetosyringone, followed by 22% and 15% on 200 and 100 µM/L respectively. In comparison, no response was recorded on the co-cultivation period of 4 days with all tested concentrations of acetosyringone as all the calli died and no regeneration was recorded. In our experiments, calli were best able to regenerate with 2 days of co-cultivation in 300 µM/L of acetosyringone (Table 4).

Calli showed proliferation of 55.5% on SMI compared to 36.7% on SMII (Table 5). Surviving calli on selection media doubled in size compared to the initial calli obtained on CIM.

After 3 to weeks on selection media, well-developed calli were transferred to two different regeneration media, RGMI (hygromycin 25 mg/L, carbenicillin 400 mg/L) and RGM II (hygromycin 25 mg/L, carbenicillin, 400 mg/L, CuO- NPs 20 mg/L) to obtain putative transgenic plants. Table 6 represents regeneration/transformation efficiency on both regeneration media. On RGM II, 52% of plantlets were regenerated from hygromycin-resistant calli; on RGMI, in contrast, 36% of resistant calli regenerated to form plantlets. As calli withstood up to 400 mg/L carbenicillin and 25 mg/L hygromycin, these concentrations were used to stop growth of *Agrobacterium* in the shoot regeneration media.

**Table 3. Effect of age of calli and infection time with *Agrobacterium* strain EHA 101 on the transformation frequency.**

| Callus age | Infection time(min) | Mean no. of calli showed growth± S.E | % of transformation |
|---|---|---|---|
| 2 weeks | 5 | 15.00± 1.01c | 30.00% d |
| | 15 | 18.34±0.55b | 36.00% c |
| | 25 | 9.70±1.9d | 19.00% e |
| 4weeks | 5 | 20.54±2.1b | 41.00% b |
| | 15 | 29.7±1.17a | 59.00% a |
| | 25 | 17.00±1.17c | 34.00% c |
| 6 weeks | 5 | 18.33±0.79b | 36.00% c |
| | 15 | 16.7±0.91c | 33.00% c |
| | 25 | 5.33±1.3e | 10.00% f |

Each value in the table is represented as mean ± SD (n = 3). Each replication consists of 20 calli. Values with the same alphabets within columns are not significantly different at ($p < 0.05$).

**Table 4. Effects of time of co-cultivation and concentration of acetosyringone on transformation efficiency.**

| Co-cultivation time | Conc. of acetosyringone (μM/L) | Mean no. of calli proliferated ± S. E | Transformation efficiency (%) |
|---|---|---|---|
| **2 days** | 100 | 6.33±0.41d | 23.00% d |
|  | 200 | 16.00±0.79 b | 53.00% b |
|  | 300 | 19.33±1.11 a | 64.00% a |
|  | 400 | 6.33±1.31d | 23.00% d |
| **3 days** | 100 | 4.67±2.11 e | 15.00% e |
|  | 200 | 6.67±0.81 d | 22.00% d |
|  | 300 | 10.00±0.54 c | 33.00% c |
|  | 400 | 3.72±1.76 e | 12.00% e |
| **4 days** | 100 | 0.00±0.00 f | 0.00% f |
|  | 200 | 0.00±0.00 f | 0.00% f |
|  | 300 | 0.00±0.00 f | 0.00% f |
|  | 400 | 0.00±0.00 f | 0.00% f |

Each value in the table is represented as mean ± SD (n = 3). Each replication consists of 20 calli. Values with the same alphabets within columns showed no significant difference at (p< 0.05).

After the regeneration of the putative transgenic plants, molecular analysis was done. Different stages of transformation of rice cv. Basmati-385 is shown in Fig 5.

## Molecular analysis

Of eight tested plants from the T0 generation, six were positive for amplification of 196 bps (Fig 6a) internal sequence of *MC1*, and 5 out of 6 plants amplified 610 bps (Fig 6b) of *hpt* gene. For the T1 generation, all 10 tested plants showed the presence of MC1 gene (Fig 6c) and nine plants tested positive for the presence of the *hpt* gene (Fig 6d). Transgenic rice lines showing PCR-positive results were further evaluated by quantitative real-time PCR for the expression of the *MC1* gene.

## Expressional analysis of *MC 1* gene

Varying levels of transgene expression were found. For example, transgenic plants of the T0 generation at tillering stage had the highest gene expression (15.41 in transgenic line 4, followed by 9.61 in transgenic line 1 and least a fold change in integrated *MC1* gene was recorded in transgenic line 2 with a value of 2.95) (Fig 7a). In contrast gene expression ranged from 6 folds change to 17 folds were noted in the T0 generation at the tillering stage. Transgenic lines 4 and 1 showed the highest gene expression level of 17.22 and 16.08, respectively. The fold change in gene expression of transgenic lines 3 and 4 was 7.08 and 6.73, respectively (Fig 7b).

The expression of *MC1* gene was also analyzed in T1 generation. Expression of the *MC1* gene revealed that the highest fold change was in transgenic line 3 i-e, 19.91 (Fig 7c). In

**Table 5. Evaluation of transformation efficiency of rice calli on selection media I & II.**

| Selection media | Mean no. of calli proliferated ± S.E | % of transformation |
|---|---|---|
| **SM I** | 16.66±1.69a | 55.5%a |
| **SM II** | 11.12± 0.34b | 36.7%b |

Each value in the table is represented as mean ± SD (n = 3). Each replication consists of 20 calli. Values with the same alphabets within columns are not significantly different at (p< 0.05).

**Table 6. Transformation frequency of hygromycin-resistant calli on regeneration medium.**

| Media | No. of hygro. resistant calli inoculated | Putative transgenic plants | Transformation frequency (%) |
|-------|------------------------------------------|----------------------------|------------------------------|
| RGM I | 25 | 09 | 36% |
| RGM II | 25 | 13 | 52% |

transgenic line 1 the fold change was recorded as 9.67. In transgenic lines 2 and 4, 2.49 and 3.06 increase in *MC1* expression was recorded compared to the internal control (Fig 7c). Maximum expression was observed at the tillering stage of T1 generation as transgenic lines 1 and 2 with a fold change of 25.65 and 22.96, respectively; for transgenic lines 3 and 4-, 15.17- and 8.81-fold change, respectively, was recorded compared to the internal control (Fig 7d).

## Bioassay to evaluate transgenic rice response against *P. oryzae*

Table 7 explains the results of the detached leaf assay from T1 generation of transgenic rice. The uppermost 3 leaves of mature were detached and inoculated with *P. oryzae*. One week post inculcation, results were recorded. Transgenic lines T1-L1, T1-L2 and T1-L3 showed the highest resistance against Blast pathogens. Meanwhile, transgenic line 4 showed moderate resistance against infected fungal pathogens. In contrast, the non-transformed rice plant was highly susceptible to fungal attack as its whole surface area of leaves became symptomatic.

## Discussion

Pathogenesis-associated (PA) proteins play a vital role against invading organisms in plant defense [30]. Despite having different PA-proteins, only plant β-1,3-glucanases and chitinases have been extensively studied [30]. Genes encoding β-1,3-glucanases and chitinases enzymes

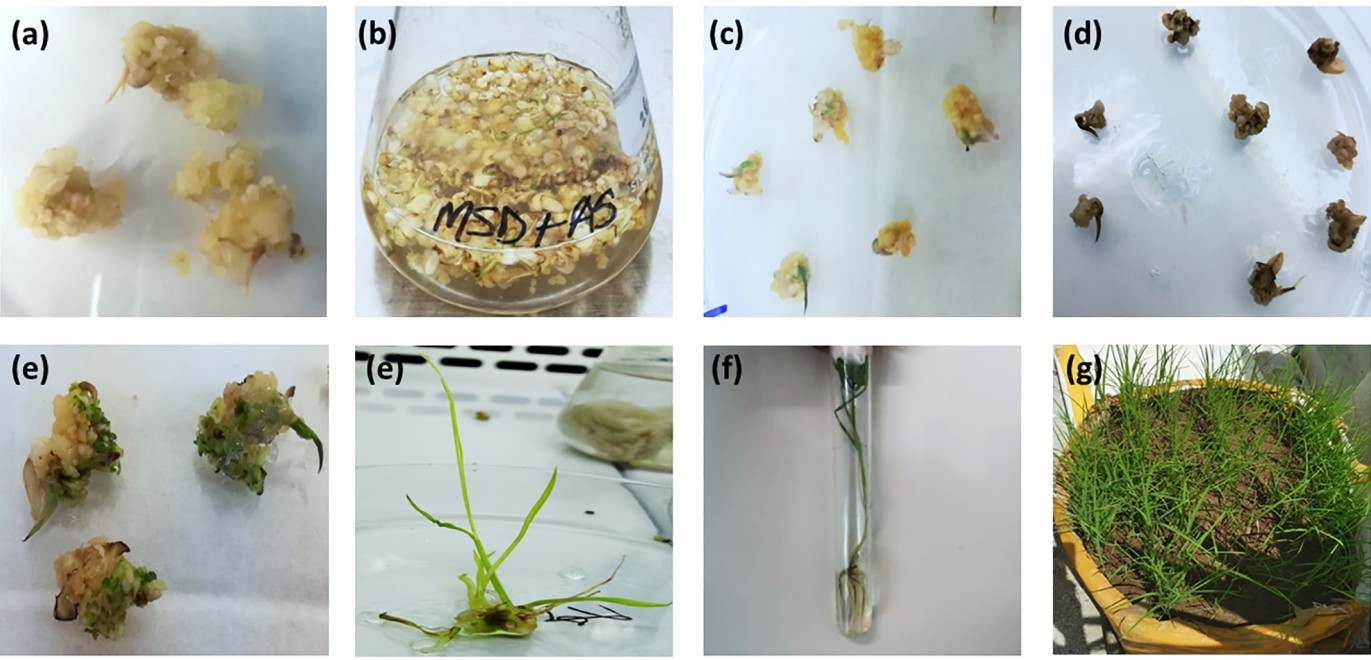

**Fig 5. Different phases of *Agrobacterium*-mediated gene transformation of Rice cv. Basmati 385. a)** Callus induction **b)** Agro-infection **c)** Co-Cultivation **d)** Selection **e)** Regeneration **f)** shooting **g)** rooting and **h)** hardening.

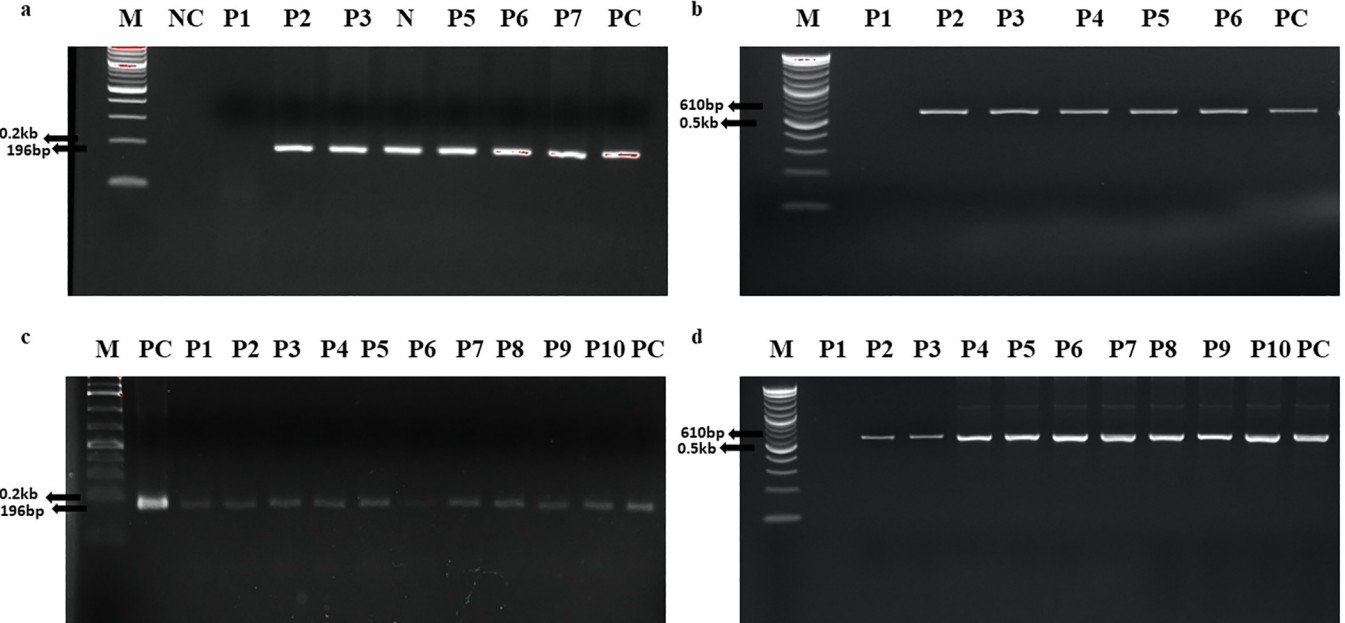

**Fig 6. (a) PCR analysis of Transgenic plants in T0 generation**. Lane 1 = Marker, Lane 2 = Non-Transgenic Plants, Lane 3–9 = Transgenic plants, Lane 10 = Positive control (Plasmid DNA) **(b) PCR detection of *hpt* gene in transgenic plants of T0 generation**. Lane 1 = Marker, Lane 2 = Non-Transgenic Plant, Lane 3–7 = Transgenic Plants, Lane 8 = Positive control (Plasmid DNA), **(c) PCR analysis of Transgenic plants in T1 generation**. Lane 1 = Marker, Lane 2 = Positive control (Plasmid DNA), Lane 3–10 = Transgenic Plants, **(d) PCR detection of *hpt* gene in transgenic plants of T1 generation**. Lane 1 = Marker, Lane 2 = Non-Transgenic Plants, Lane 3–11 = Transgenic Plants, Lane 12 = Positive control (Plasmid DNA).

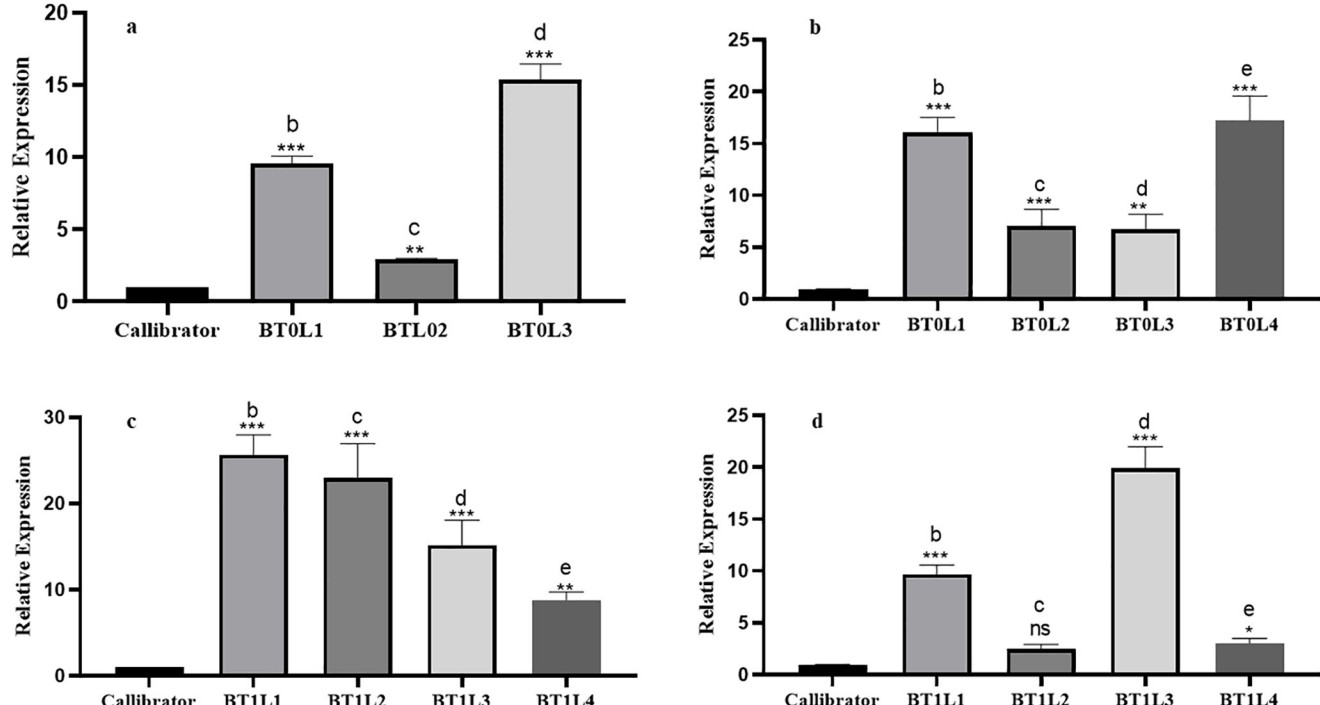

**Fig 7. (a) Relative Expression of *MC1* in regenerated stage at T0 generation, (b) Relative Expression of *MC1* in tillering stage at T0 generation, (c) Relative Expression of *MC1* in seedling stage at T1 generation, (d) Relative Expression of *MC1* in tillering stage at T1 generation.**

**Table 7. Functional validation of transgenic plants by leaf detached assay.**

| Rice Variety | Response against *P.oryzae* | | | |
|:---:|:---:|:---:|:---:|:---:|
| | Lesion size± S.E(cm) | % of Blast | Disease Score | Response |
| T1-L1 | 0.55±1.2 c | 6.9 | 1 | Highly Resistant |
| T1-L2 | 1.21±0.2 c | 8.0 | 1 | Highly Resistant |
| T1-L3 | 0.78±1.0 c | 7.5 | 1 | Highly Resistant |
| T1-L4 | 3.54±0.41b | 13.2 | 2 | Moderately Resistant |
| T0 | 8.66±1.1 a | 100 | 5 | Highly Susceptible |
| Control | 0.00 0.00 c | 0 | 0 | Nil |

Values followed by the same letter are non-significant at α 0.01, n = 3 replicates. L1-L3 represent transgenic lines

have also been cloned from the various plant [31]. Currently, determining the molecular mechanism from pathogen identification for the expression analysis of such genes is of prime interest. Several plant sources have been altered using PA genes to increase disease resistance [32]. According to the literature, the expression of PA genes and disease resistance are positively correlated [32]. Over half of the world's population depends on rice (*Oryza sativa* L.) as a staple food [33]. One of the leading causes of biotic stress in rice, which results in significant productivity losses, is fungus infections. Rice blast is currently the most threatening rice disease in the world. It poses a concern to food security because of its capacity to decrease yields approximately in all geographic distributions. At every stage of development, the pathogen (*Magnaporthe oryzae*) attacks rice plants. It damages them, causing blast symptoms on all parts of the plant, including the collar, leaves, neck, panicle, node and grain. Each year, the fungus damages enough rice to feed approximately 60 million people, and the occurrence of blasts causes 50% of the rice production to be lost in the field. Regarding geographic coverage, rice blast is estimated to reduce significant productivity and result in a loss of millions of dollars. Although there are resistant varieties, rice blast is present wherever rice is farmed and has never been entirely eradicated, although the resistance wears off in a short period. Conventionally, rice disease management is carried out by breeding methods which are laborious and time-consuming processes. On the other hand, targeted genome editing has the potential to produce disease-resistant variants more quickly [34, 35].

Different factors can affect Agrobacterium transformation's efficacy, which is why multiple parameters are needed to optimize to achieve maximum transformation efficiency [36]. Additional parameters like age of calli, antibiotic selection, the concentration of acetosyringone, time of agro-infection and co-cultivation period are also important factors of optimization. In our study, hygromycin concentration of 25mg/L was more effective than other concentrations. Likewise, in the case of indica rice, Yaqoob et al. [37] used 25 mg/l of hygromycin for selection and regeneration purposes. The callus' age is crucial to achieving a high transformation frequency [36]. In our investigation, calli 2, 3, and 4 weeks old were removed and infected for 5, 15, and 25 minutes with an agrobacterium strain. The 3 week old calli that had been exposed to the agro strain for 15 minutes showed the best response. The same outcomes were seen by Rao et al. [38] in indica rice embryogenic calli at three and four weeks old. Optimal immersion time is necessary because *Agrobacterium* overgrowth during selection could also result from extended immersion time. Hence the ideal immersion time should be followed. *Agrobacterium* overgrowth can make the sub-culturing more tedious and challenging. According to several investigations, the resistant monocots, in particular, required phenolic inducers such as acetosyringone during gene transfer [39]. Therefore, it's essential to utilize acetosyringone at the right concentration to avoid low transformation [26]. Depending on the plant type, in most of

the reported studies, acetosyringone ranges from 20 to 200 M [40]. However, in the current investigation, 300 M of acetosyringone was found to be efficient, with a more than 60% regeneration rate. The findings of the current investigation are supported by the outcomes reported by Tan et al. [41]. The duration of the co-cultivation period is another crucial element that significantly affected the effectiveness of the transformation. Depending on the plant type, co-cultivation might range from a few hours to a few days [42]. Another critical factor is the co-cultivation period which showed a substantial effect on transformation efficiency. The co-cultivation time varies from hours to a few days, depending on plant species [42]. In the present research, we have found that two days of co-cultivation time was optimum for transformation, which is in the range of normal reported co-cultivation duration for rice, being two to three days [43].

Several genes that encode antifungal proteins have been effectively transgenetically expressed against various plant pathogenic fungi [44]. Amongst the pathogenesis-related proteins, chitinases represent the class of chitinolytic enzymes. Chitinases isolated from different sources show multiple ranges of antifungal activities in vitro. Mainly, chitinases class I is reported for the highest antifungal activity, which might be due to the chitin-binding domain presence [45]. Chitin-degrading enzymes in plants are plentiful and vastly diverse, including those in maize [46, 47]. A maize chitinase that has been related to fungal resistance is *chitinase* 2. Maize *chitinase* 2 has been expressed by inducing plants with *A. flavus* [48] and *Fusarium graminearum* Schwabe [49].

The goal of the current study was to clone and integrate a synthetic maize *chitinase* 1 gene using Agrobacterium-mediated transformation to create the transgenic rice variety (Basmati-385). Molecular analysis of putative transgenic lines in the current study was performed using PCR which confirmed the integration of *MC1* gene expression cassette into the rice genomes. Progeny of selected transgenic plants showed variable expression levels of *MC1* gene in qRT-PCR analysis. Expressional analysis was performed at the seedling and tillering stage on both generations (T0 and T1). *MC1* gene showed higher expression than actin at seedling and tillering stages. The maximum gene expression compared to internal control was recorded in the seedling stage of T1 generation. It is reported that the prominent heterogeneity in transgene expression is frequently observed when the transgene construct is the same in different transgenic lines [17, 50, 51]. It is mainly due to the position effect [52], transgene copy [53] and various epigenetic silencing phenomena [54]. Bandopadhyay et al. [55] explained that high gene expression is linked with a single-site transgene insertion, whereas multiple sites transgene insertions may result in gene silencing. In addition, methylation of the promoter may also cause gene silencing, as extended methylation affects the gene expression level [56]. Literature reports that a higher number of *cis*-elements in their promoters is also associated with higher expression [57].

One of the most devastating diseases impacting the world's rice production is rice blasts [58]. *M. oryzae* directly enters the cell membranes, causing mycelia to proliferate inside the cell and eventually kill the cell. Mycelia are believed to transfer to the neighboring cell before cell death, possibly by plasmodesmata. The growth of the fungus also hampers water and mineral movement in the plant's vascular system. The fungus generates thousands of spores on conidiophores that emerge from stomata after entering the plant, and air currents can spread these spores to neighboring rice plants for prospective infection. This fungus is highly adaptable and can cause infection in its host at any stage of the growth cycle [59, 60]. Functional validation of the inserted gene was performed on transgenic plants based on the findings of the expressional study of MC1 by detached leaf bioassay with *P. oryzae*, the causative agent of Blast disease, to check the plant antifungal potential. Results have revealed significantly increased resistance against the tested pathogen. This resistance can be explained in terms of effector-triggered and

pattern-triggered immunity. Pathogen-associated molecular patterns typically involve microbial or pathogen structures like lipopolysaccharides, flagellins, glucans and chitins recognized by specific plant receptors known as pattern recognition receptors that promote the activation of pattern-triggered immunity [61]. The variability in disease resistance of tested lines can be described as variable gene expression, as 3 tested transgenic lines were highly resistant, and one was moderately resistant. Our results follow previous findings in which the transgenic plants significantly reduced lesion area because of better penetration resistance and constrained pathogen lesion growth at infection sites [62].

The development of disease signs and higher resistance to infection by the pathogenic fungus *Rhizoctonia solani* were both observed in transgenic plants in published studies [63]. Various plants till now have been transformed using chitinase genes that exhibit improved disease resistance [64]. Furthermore, numerous publications state that non-plant chitinases have been introduced to model plants to assess their utility in disease resistance [65]. For example, the mycoparasitic fungus *Trichoderma harzianum's* chitinase gene has been transferred to tobacco and potato plants [66]. The transgenic plants displayed improved resistance to several fungi, such as *Alternaria solani*, *Alternaria alternata*, *Rhizoctonia solani* and *Botrytis cinerea* [66]. In another study, when exposed to pathogenic fungi, a different fungal chitinase gene from *Rhizopus oligosporus* suppressed disease symptoms in transgenic tobacco plants [67]. Literature also reports that tobacco transformed with the chitinase gene of autograph californica multiple nucleopolyhedro viruses showed enhanced disease resistance in approximately 60% of plants [68].

In addition, the *MC1* integration and expression exhibited no harmful effects on the transgenic plants and high-level resistance against blast disease, consistent with the results explaining the nontoxic effects of chitinases on host plants [69]. Therefore, chitinases restrict fungus growth by breaking down their cell walls; they also stop hyphae from growing and bud tubes from lengthening [70]. Additionally, it breaks down chitin into chitin oligosaccharides, which plants use as elicitors to activate their innate immunity [71] and subsequent host defensive mechanisms [69].

## Conclusion

This is the first study to report the antifungal activity of maize *chitinase* 1 gene, expressed in rice cv. Basmati 385. The results showed that the constitutive expression of maize class I chitinase in rice resulted in enhanced antifungal resistance against *P. oryzae*. As Blast is one of the most destructive fungal diseases that cause huge yield loss per year. This work and further research will possibly add the maize *chitinase* 1 gene to the list of chitinases that are useful in genetic manipulation strategies for plants to develop resistance against phytopathogenic fungi.

## Supporting information

**S1 Raw images.**
(PPTX)

## Acknowledgments

We thank Dr. David W. Wright, Iowa State University, Ames, Iowa, USA, for providing vector pDW3586.

## Author Contributions

**Conceptualization:** Nyla Jabeen.

**Data curation:** Sadaf Anwaar, Saima Shafique, Samra Irum.

**Formal analysis:** Sadaf Anwaar, Saima Shafique, Samra Irum, Mark L. Gleason.

**Funding acquisition:** Nyla Jabeen.

**Investigation:** Sadaf Anwaar.

**Methodology:** Siffat Ullah Khan.

**Resources:** Nyla Jabeen, Mark L. Gleason.

**Software:** Sadaf Anwaar, Siffat Ullah Khan, Ateeq Tahir, Mark L. Gleason.

**Supervision:** Nyla Jabeen, Khawaja Shafique Ahmad, Mark L. Gleason.

**Validation:** Saima Shafique, Hammad Ismail, Siffat Ullah Khan, Ateeq Tahir, Nasir Mehmood.

**Visualization:** Hammad Ismail, Siffat Ullah Khan, Nasir Mehmood.

**Writing – original draft:** Sadaf Anwaar, Nyla Jabeen.

**Writing – review & editing:** Nyla Jabeen, Khawaja Shafique Ahmad, Mark L. Gleason.

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
