## [Decision Letter · Decision Letter 0]

23 May 2022

PONE-D-21-36584Cloning of Maize Chitinase 1 gene and its expression in genetically transformed Rice: augmented resistance against pathogenic fungiPLOS ONE

Dear Dr. Ahmad,

Thank you for submitting your manuscript to PLOS ONE. After careful consideration, we feel that it has merit but does not fully meet PLOS ONE’s publication criteria as it currently stands. Therefore, we invite you to submit a revised version of the manuscript that addresses the points raised during the review process.

ACADEMIC EDITOR: 

Please give pointwise replies to the comments raised by the reviewers

Please ensure to have the language editing and check for typo errors

Discussion section is invariable need to improve

We look forward to receiving your revised manuscript.

Kind regards,

Maganti Sheshu Madhav, Ph.D.

Academic Editor

PLOS ONE

“We are thankful to Higher Education Commission Pakistan, for providing funding to conduct the current research. No additional external funding was received for this study.”

5. Please amend your list of authors on the manuscript to ensure that each author is linked to an affiliation. Authors’ affiliations should reflect the institution where the work was done (if authors moved subsequently, you can also list the new affiliation stating “current affiliation:….” as necessary).

Reviewers' comments:

Reviewer's Responses to Questions

**Comments to the Author**

1. Is the manuscript technically sound, and do the data support the conclusions?

Reviewer #1: Partly

Reviewer #2: No

2. Has the statistical analysis been performed appropriately and rigorously? 

Reviewer #1: I Don't Know

Reviewer #2: No

3. Have the authors made all data underlying the findings in their manuscript fully available?

Reviewer #1: Yes

Reviewer #2: Yes

4. Is the manuscript presented in an intelligible fashion and written in standard English?

Reviewer #1: No

Reviewer #2: No

5. Review Comments to the Author

Reviewer #1: 1. Language editing and check for typo errors is needed.

2. In line 122: are pSDF1 and pDW3586 same vectors?

3. Line 133-134: Whether pSDF1 and pSDF2 plasmids are of same size? How can you get 10 kb and 0.9 kb DNA size in both the vectors when restriction digested with Spe I and Bam HI ? pSDFF1 is pDW3586 while pSDF2 is pUC57 with MC1.

4. Please write the correct name of Agrobacterium strain. It is EHA 101 not the EH 101.

5. Fig. 2: Please show properly the T-DNA region. All components should be within LB and RB regions.

6. Line 185: Is it table 1 or 3.6

7. In Table 1, for infection formulation pH 7.0 was used which is not common for Agrobacterium transformation experiments. pH 5.2 is usually preferred.

8. Line 216-220. Denaturation in PCR at 72oC is not correct. Please verify.

9. Line 260-262: Whether plasmid pDW3586 has two sizes- 13.5 kb and 4 kb? In D1-D3 lanes 9.1 kb and 6 kb bands were inappropriate. Moreover, in D2, D3 lanes bands are absent. Mostly the gel image are not matching with the text given in results. Provide an appropriate gel image (3b).

10. Fig5. Replace with good image. Also add the rooting and hardening stage images.

11. Comment 16: In table 6, just mention as T0 or T1 lines. No need to write name of rice variety.

12. Line 419: Some text is missing in leaf detachment assay

13. Whole plant assay must be conducted along with leaf detachment assay and the results should be included.

14. Lines 424-427: Sentences are repeated from abstract and introduction sections.

15. Discussion should be improved a lot. Nowhere in the entire manuscript was mentioned regarding the significance of blast disease. What is the importance of this work? Most of the work is w.r.t. standardizing the protocol for Basmati variety. The discussion should have focused on chitinase gene and blast disease.

16. In the sentences such as “High gene expression level was explained by [69]”, authors name should be written along with the number for citation.

17. Instead of using the term parent vector, please use standard terms such as cloning vector, expression vector and binary vectors or simply a vector.

Reviewer #2: The article is of scientific interest but the article needs thorough revision. It is better to get it edited using an expert in English language. The MS is too elaborative. Fig-7 does not represent the differences of control and transgenic plants inoculated with P. oryzae. The Figure 7(a) it seems all the plants are suffering from nutrient deficiency. Similarly, Please check the Fig7. (b) there are huge differences among all the three plants as per as the nutrient supply is concerned. The first criteria of blast infection is the application of heavy dose of nitrogen to the plant which is not seen in your experiment. The species of Agrobacterium is not mentioned any where. In line 233 the authors have written "Total RNA extraction was done by method reported by [84] with slight modification": this type of sentence should be avoided. The sentence should be written as "reported by Liu et.al. [84]". Kindly modify the MS thoroughly checking even the technicality and the language. In Fig 3 why you have used two different background for the gel photography of 3.b & d?

6. PLOS authors have the option to publish the peer review history of their article (what does this mean?). If published, this will include your full peer review and any attached files.

Reviewer #1: No

Reviewer #2: **Yes: **Dr. Arup Kumar Mukherjee

---

## [Author Response · Author response to Decision Letter 0]

17 Nov 2022

Response to Academic Editor: 

Please give pointwise replies to the comments raised by the reviewers

Answer: All comments are replied

Please ensure to have the language editing and check for typo errors

Answer: Manuscript has been revised and proofread completely to remove all language, grammar and typing errors.

Discussion section is invariable need to improve

Answer: Discussion part has been revised completely as per reviewers guidelines

Response to Journal Requirements:

Answer: Changes has been incorporated. 

Answer: Please add the following grant number “HEC-NRPU-1854”

Answer: Please update our Funding Statement as following 

“We are thankful to Higher Education Commission Pakistan, for providing funding to conduct the current research under grant number HEC-NRPU-1854. No additional external funding was received for this study. The funding agency had no role in study design, data collection and analysis, decision to publish, or preparation of the manuscript.”

4. PLOS ONE now requires that authors provide the original uncropped and unadjusted images underlying all blot or gel results reported in a submission’s figures or Supporting Information files. In your cover letter, please note whether your blot/gel image data are in Supporting Information or posted at a public data repository, provide the repository URL if relevant, and provide specific details as to which raw blot/gel images, if any, are not available. Email us at plosone@plos.org if you have any questions.

Answer: Please add the data as supporting information. 

5. Please amend your list of authors on the manuscript to ensure that each author is linked to an affiliation. Authors’ affiliations should reflect the institution where the work was done (if authors moved subsequently, you can also list the new affiliation stating “current affiliation:….” as necessary).

Answer: List of authors is linked with affiliation.  

Response to Reviewers Comments

Reviewer #1

1. Language editing and check for typo errors is needed. Manuscript has been revised and proofread completely to remove all such errors.

2. In line 122: are pSDF1 and pDW3586 same vectors? Yes, you are right this is the same vector and it has been explained in revised manuscript. 

3. Line 133-134: Whether pSDF1 and pSDF2 plasmids are of same size? How can you get 10 kb and 0.9 kb DNA size in both the vectors when restriction digested with SpeI and Bam HI? pSDF1 is pDW3586 while pSDF2 is pUC57 with MC1. 10Kb fragment was produced after restriction digestion of pSDF1 while 0.9Kb fragment was produced after restriction digestion of pSDF2. Statement has been corrected in the revised manuscript. 

4. Please write the correct name of Agrobacterium strain. It is EHA 101 not the EH 101. Corrections has been made in the revised manuscript. 

5. Fig. 2: Please show properly the T-DNA region. All components should be within LB and RB regions. Changes has been incorporated in the revised figure. 

6. Line 185: Is it table 1 or 3.6 Its table 1. Thanks for pointing out. Correction has been made in the revised manuscript. 

7. In Table 1, for infection formulation pH 7.0 was used which is not common for Agrobacterium transformation experiments. pH 5.2 is usually preferred. Yes, you are right, in our experiments, pH was optimized as 5.8 and correction has been made in the revised manuscript. 

8. Line 216-220. Denaturation in PCR at 72oC is not correct. Please verify. Correction has been made in the revised manuscript. 

9. Line 260-262: Whether plasmid pDW3586 has two sizes- 13.5 kb and 4 kb? In D1-D3 lanes 9.1 kb and 6 kb bands were inappropriate. Moreover, in D2, D3 lanes bands are absent. Mostly the gel images are not matching with the text given in results. Provide an appropriate gel image (3b). It was typing mistake. The actual size is 17.34kb. Correction has been made in the manuscript. D1, D2 and D3 represents different time of restriction incubation and has been explained in the revised manuscript. 

10. Fig5. Replace with good image. Also add the rooting and hardening stage images. Changes has been incorporated as suggested.

11. In table 7, just mention as T0 or T1 lines. No need to write name of rice variety. Changes has been incorporated as suggested. 

12. Line 419: Some text is missing in leaf detachment assay Correction has been made in the revised manuscript. 

13. Whole plant assay must be conducted along with leaf detachment assay and the results should be included. Thank you for your positive suggestion. Infect due to limitation of T1 generation plants, we only focused on the leaf detachment assay. We used leaf detachment assay because in literature it has been used as standard assay in multiple studies. 

14. Lines 424-427: Sentences are repeated from abstract and introduction sections. Statements has been replaced in the revised version. 

15. Discussion should be improved a lot. Nowhere in the entire manuscript was mentioned regarding the significance of blast disease. What is the importance of this work? Most of the work is w.r.t. standardizing the protocol for Basmati variety. The discussion should have focused on chitinase gene and blast disease. Discussion has been revised completely as per suggestions. 

16. In the sentences such as “High gene expression level was explained by [69]”, authors name should be written along with the number for citation. Changes has been incorporated in the revised manuscript.

17. Instead of using the term parent vector, please use standard terms such as cloning vector, expression vector and binary vectors or simply a vector. Changes has been incorporated in the revised manuscript. 

Reviewer #2:

1. The article is of scientific interest but the article needs thorough revision. It is better to get it edited using an expert in English language. The MS is too elaborative. Manuscript has been revised and proofread completely to remove all such errors.

2. Fig-7 does not represent the differences of control and transgenic plants inoculated with P. oryzae. The Figure 7(a) it seems all the plants are suffering from nutrient deficiency. Similarly, please check the Fig7. (b) there are huge differences among all the three plants as per as the nutrient supply is concerned. Thank you for your suggestion. Figure 7 represents the relative gene expression of MC1 gene in T0 and T1 generation which was quantified by real time PCR. The difference in bar represents the gene expression relative to control 

3. The first criteria of blast infection is the application of heavy dose of nitrogen to the plant which is not seen in your experiment. Thank you for your insight review. Infect our main objective of the research was cloning of MC1 gene and its expression analysis so we limited our applications only on transgenic line in comparison of non-transgenic lines. 

4. The species of Agrobacterium is not mentioned anywhere. Specie information has been added in the revised manuscript. 

5. In line 233 the authors have written "Total RNA extraction was done by method reported by [84] with slight modification": this type of sentence should be avoided. The sentence should be written as "reported by Liu et.al. [84]". 

Changes has been incorporated in the revised manuscript.

6. Kindly modify the MS thoroughly checking even the technicality and the language. Manuscript has been revised completely to remove all such errors.

7. In Fig 3 why you have used two different backgrounds for the gel photography of 3.b & d? Thank you for your time and positive feedback on our manuscript. The gel images were observed using two different instruments and we intended to provide the original images in the manuscript that’s why we added them as it.

---

## [Decision Letter · Decision Letter 1]

6 Jan 2023

PONE-D-21-36584R1Cloning of Maize Chitinase 1 gene and its expression in genetically transformed Rice: augmented resistance against pathogenic fungiPLOS ONE

Dear Dr. Ahemad,

Thank you for submitting your manuscript to PLOS ONE. After careful consideration, we feel that it has merit but does not fully meet PLOS ONE’s publication criteria as it currently stands. Therefore, we invite you to submit a revised version of the manuscript that addresses the points raised during the review process.

Please submit your revised manuscript by  14th Jan 2023 .

If you will need more time than this to complete your revisions, please reply to this message or contact the journal office at plosone@plos.org. Please include the following items when submitting your revised manuscript:

We look forward to receiving your revised manuscript.

Kind regards,

Maganti Sheshu Madhav, Ph.D.

Academic Editor

PLOS ONE

Journal Requirements:

Additional Editor Comments:

Authors have addressed the concerns raised by the reviewers. But I suggest authors to improve the language of the paper by taking help of the professional English editing services.

The use of mycelial plugs from 48 hr old culture to inoculate for testing disease resistance need to substantiate...

Reviewers' comments:

Reviewer's Responses to Questions

**Comments to the Author**

1. If the authors have adequately addressed your comments raised in a previous round of review and you feel that this manuscript is now acceptable for publication, you may indicate that here to bypass the “Comments to the Author” section, enter your conflict of interest statement in the “Confidential to Editor” section, and submit your "Accept" recommendation.

Reviewer #3: All comments have been addressed

Reviewer #4: (No Response)

2. Is the manuscript technically sound, and do the data support the conclusions?

Reviewer #3: Yes

Reviewer #4: No

3. Has the statistical analysis been performed appropriately and rigorously? 

Reviewer #3: Yes

Reviewer #4: Yes

4. Have the authors made all data underlying the findings in their manuscript fully available?

Reviewer #3: Yes

Reviewer #4: Yes

5. Is the manuscript presented in an intelligible fashion and written in standard English?

Reviewer #3: Yes

Reviewer #4: No

6. Review Comments to the Author

Reviewer #3: The authors have incorporated all the modifications as suggested. But till then I suggest please check the spellings once more.

Reviewer #4: I have gone through the comments of earlier reviewers. While the authors have tried to address the concerns raised by the reviewers, I still believe that the authors should improve the language of the paper by taking help of the professional English editing services.

I have noted a gross methodological error committed by the authors while screening the resistance of the trasngenics expressing the chitinase gene. The use of the mycelial plugs from 48 hr old culture to inoculate rice leaves for testing their resistance is absolutely incorrect because the mycelial strands cannot penetrate and infect the aerial plant surfaces of the rice. Probably the authors are not aware about the infection biology of the blast pathogen. The infection of rice leaves invariably occurs through the conidial inoculum, which is applied to the leaves as spore suspension or as spore droplets. Infection begins when spores germinate on the leaf surface and form appressoria at the germ tube tips. Appressoria become pressurized and melanized. When tightly annealed, hydrostatic turgor acts on a penetration peg at the appressorial base, forcing it to penetrate the cuticle. It is therefore necessary to apply a conidial suspension to leaves to evaluate their resistance to blast. I am providing the references of a few papers, which report on evaluation on the resistance of transgenics expressing chitinase or other defence response genes against rice blast; in all these studies conidial suspensions have been applied to test the resistance of rice plants against leaf blast.

i. Nishizawa et al. 1999. Enhanced resistance to blast (Magnaporthe grisea) in transgenic Japonica rice by constitutive expression of rice chitinase. Theor Appl Genet. 99(3-4):383-90.

ii. Qian et al. 2014. Enhanced resistance to blast fungus in rice (Oryza sativa L.) by expressing the ribosome-inactivating protein α-momorcharin. Plant Science : an International Journal of Experimental Plant Biology. 217-218:1-7. DOI: 10.1016/j.plantsci.2013.11.012.

iii. Pokhrel et al. 2021. Transgenic Rice Expressing Isoflavone Synthase Gene from Soybean Shows Resistance Against Blast Fungus (Magnaporthe oryzae). Plant Dis. 105(10):3141-3146.

The mycelial plugs can be used to test the resistance of rice to other pathogens like Rhizoctonia solani which elaborate specialized pentation structures from mycelial strands to penetrate host surface. The same procedure cannot be applied against rice blast due to peculiar infection biology of the rice blast pathogen. The references no.12 and 28 provided by the authors in material and methods section also reported the use the conidial spore suspensions to test the resistance of the respective host species. I therefore wonder that on what premise the authors have used the mycelial plugs to test the resistance of transgenics in their study. The use of mycelial plugs for testing the resistance against leaf blast has vitiated the authenticity of the results of the present study.

I do not recommend the paper for publication in PLOS.

7. PLOS authors have the option to publish the peer review history of their article (what does this mean?). If published, this will include your full peer review and any attached files.

Reviewer #3: **Yes: **Arup Kumar Mukherjee

Reviewer #4: No

---

## [Author Response · Author response to Decision Letter 1]

16 Jan 2023

Journal Requirements:

Please review your reference list to ensure that it is complete and correct. If you have cited papers that have been retracted, please include the rationale for doing so in the manuscript text or remove these references and replace them with relevant current references. Any changes to the reference list should be mentioned in the rebuttal letter that accompanies your revised manuscript. If you need to cite a retracted article, indicate the article’s retracted status in the References list and also include a citation and full reference for the retraction notice.

Answer: All references has been cross checked and corrections has been made in the revised version

Additional Editor Comments:

Authors have addressed the concerns raised by the reviewers. But I suggest authors to improve the language of the paper by taking help of the professional English editing services.

Answer: Manuscript has been revised and proofread completely by American native language speaking person (Letter attached) to remove all language, grammar and typing errors.

The use of mycelial plugs from 48 hr old culture to inoculate for testing disease resistance need to substantiate...

Answer: We agree with the reviewer comments and experiment validation has been added in the revised version. 

Review Comments to the Author

Reviewer #3: The authors have incorporated all the modifications as suggested. But till then I suggest please check the spellings once more.

Answer: Thank you for your time and feedback. Manuscript has been revised and proofread completely to remove all language, grammar and typing errors.

Reviewer #4: I have gone through the comments of earlier reviewers. While the authors have tried to address the concerns raised by the reviewers, I still believe that the authors should improve the language of the paper by taking help of the professional English editing services.

Answer: Manuscript has been revised and proofread completely to remove all language, grammar and typing errors.

I have noted a gross methodological error committed by the authors while screening the resistance of the trasngenics expressing the chitinase gene. The use of the mycelial plugs from 48 hr old culture to inoculate rice leaves for testing their resistance is absolutely incorrect because the mycelial strands cannot penetrate and infect the aerial plant surfaces of the rice. Probably the authors are not aware about the infection biology of the blast pathogen. The infection of rice leaves invariably occurs through the conidial inoculum, which is applied to the leaves as spore suspension or as spore droplets. Infection begins when spores germinate on the leaf surface and form appressoria at the germ tube tips. Appressoria become pressurized and melanized. When tightly annealed, hydrostatic turgor acts on a penetration peg at the appressorial base, forcing it to penetrate the cuticle. It is therefore necessary to apply a conidial suspension to leaves to evaluate their resistance to blast. I am providing the references of a few papers, which report on evaluation on the resistance of transgenics expressing chitinase or other defence response genes against rice blast; in all these studies conidial suspensions have been applied to test the resistance of rice plants against leaf blast.

Answer: Yes, you are right. We also used the spores suspension for testing the disease resistance. Initially we added the summarized version of methodologically in which we unintendedly, missed some important steps. Thanks for your insightful review to highlight the importance and mechanism of the bioassay. The detail of methodology has been added in the revised manuscript. This revised methodology is SOP of our laboratory, as in one of the previous studies we also utilized the same methodology in which spores were utilized.

ANWAAR, SADAF, et al. "EVALUATION OF PAKISTANI RICE CULTIVARS FOR APPARENT INFECTION RATE AGAINST FUNGAL PATHOGEN Pyricularia oryzae." PLANT CELL BIOTECHNOLOGY AND MOLECULAR BIOLOGY (2021): 29-37.

---

## [Decision Letter · Decision Letter 2]

20 Mar 2023

PONE-D-21-36584R2Cloning of maize chitinase 1 gene and its expression in genetically transformed rice to confer resistance against rice blast caused by Pyricularia oryzaePLOS ONE

Dear Dr. Ahmad,

Thank you for submitting your manuscript to PLOS ONE. After careful consideration, we feel that it has merit but does not fully meet PLOS ONE’s publication criteria as it currently stands. Therefore, we invite you to submit a revised version of the manuscript that addresses the points raised during the review process.

We look forward to receiving your revised manuscript.

Kind regards,

Maganti Sheshu Madhav, Ph.D.

Academic Editor

PLOS ONE

Journal Requirements:

Additional Editor Comments:

The article is of scientific interest, but the article needs still revision. It is better to get it edited using an expert in English language.

The manuscript is too elaborative. Figures need to be revised. Discussion needs to be improved a lot. Please go through several comments of reviewers and reply suitably.

Reviewers' comments:

Reviewer's Responses to Questions

**Comments to the Author**

1. If the authors have adequately addressed your comments raised in a previous round of review and you feel that this manuscript is now acceptable for publication, you may indicate that here to bypass the “Comments to the Author” section, enter your conflict of interest statement in the “Confidential to Editor” section, and submit your "Accept" recommendation.

Reviewer #3: All comments have been addressed

Reviewer #4: (No Response)

2. Is the manuscript technically sound, and do the data support the conclusions?

Reviewer #3: Yes

Reviewer #4: No

3. Has the statistical analysis been performed appropriately and rigorously? 

Reviewer #3: Yes

Reviewer #4: Yes

4. Have the authors made all data underlying the findings in their manuscript fully available?

Reviewer #3: Yes

Reviewer #4: (No Response)

5. Is the manuscript presented in an intelligible fashion and written in standard English?

Reviewer #3: (No Response)

Reviewer #4: No

6. Review Comments to the Author

Reviewer #3: The authors have answered all the queries asked by the reviewers. My only concern is about the quality of photographs especially the gel photographs.

Reviewer #4: As has been the case with many of the queries raised by other reviewers, in the instant case also the authors knowing well that they have committed a fatal methodological error in screening of their transgenic lines, they have changed their stance and now claiming that they have the used the other screening method suggested by me to screen the transgenic lines. Their assertion is unbelievable and cannot be trusted.

7. PLOS authors have the option to publish the peer review history of their article (what does this mean?). If published, this will include your full peer review and any attached files.

Reviewer #3: **Yes: **Arup Kumar Mukherjee

Reviewer #4: No

---

## [Author Response · Author response to Decision Letter 2]

24 Jul 2023

Journal Requirements:

Please review your reference list to ensure that it is complete and correct. If you have cited papers that have been retracted, please include the rationale for doing so in the manuscript text or remove these references and replace them with relevant current references. Any changes to the reference list should be mentioned in the rebuttal letter that accompanies your revised manuscript. If you need to cite a retracted article, indicate the article’s retracted status in the References list and also include a citation and full reference for the retraction notice.

Answer: Reference list has been checked carefully, it is complete and moreover no paper has been cited which is retracted.

Additional Editor Comments:

The article is of scientific interest, but the article needs still revision. 

It is better to get it edited using an expert in English language. 

The manuscript is too elaborative. Figures need to be revised. Discussion needs to be improved a lot. Please go through several comments of reviewers and reply suitably

Answer: Manuscript has been revised 

 

Review Comments to the Author

Reviewer #3: The authors have answered all the queries asked by the reviewers. My only concern is about the quality of photographs especially the gel photographs.

Answer: Thank you for your time and feedback. Quality of manuscript has been improved in the revised manuscript. 

Reviewer #4: As has been the case with many of the queries raised by other reviewers, in the instant case also the authors knowing well that they have committed a fatal methodological error in screening of their transgenic lines, they have changed their stance and now claiming that they have the used the other screening method suggested by me to screen the transgenic lines. Their assertion is unbelievable and cannot be trusted.

Answer: Thank you for your time and concern regarding methodology. Initially, revision was requested on leaf detached assay and we are thankful for the reviewer for identifying missing information. We have just added the missing information and provided the complete methodology. Additionally, we would like to add that using spore suspension is a routine methodology of our laboratory and it is our standardized SOP which can also be confirmed from our other publication “ANWAAR, SADAF, et al. "EVALUATION OF PAKISTANI RICE CULTIVARS FOR APPARENT INFECTION RATE AGAINST FUNGAL PATHOGEN Pyricularia oryzae." PLANT CELL BIOTECHNOLOGY AND MOLECULAR BIOLOGY (2021): 29-37.” Moreover, the reference we cited in the relevant section also support our methodology for using spore suspension. We just used the summary of methodology and cited the reference which ultimately means that we have followed the same method as mentioned in the reference.

---

## [Editor Report · Decision Letter 3]

10 Sep 2023

Cloning of maize chitinase 1 gene and its expression in genetically transformed rice to confer resistance against rice blast caused by Pyricularia oryzae

PONE-D-21-36584R3

Dear Dr. Ahmad,

We’re pleased to inform you that your manuscript has been judged scientifically suitable for publication and will be formally accepted for publication once it meets all outstanding technical requirements.

Kind regards,

Maganti Sheshu Madhav, Ph.D.

Academic Editor

PLOS ONE

Additional Editor Comments (optional):

The authors have adequately addressed your comments raised in a previous round of review and I feel that this manuscript is now acceptable for publication.
---

## [Editor Report · Acceptance letter]

13 Sep 2023

PONE-D-21-36584R3 

Cloning of maize *chitinase* 1 gene and its expression in genetically transformed rice to confer resistance against rice blast caused by *Pyricularia oryzae*

Dear Dr. Ahmad:

I'm pleased to inform you that your manuscript has been deemed suitable for publication in PLOS ONE. Congratulations! Your manuscript is now with our production department. 

Kind regards, 

on behalf of

Dr. Maganti Sheshu Madhav 

Academic Editor

PLOS ONE